# In Vitro and Ex Vivo Investigation of the Antibacterial Effects of Methylene Blue against Methicillin-Resistant *Staphylococcus aureus*

**DOI:** 10.3390/ph17020241

**Published:** 2024-02-13

**Authors:** Deniz Gazel, Mehmet Erinmez, Gönenç Çalışkantürk, Khandakar A. S. M. Saadat

**Affiliations:** 1Department of Medical Microbiology, Faculty of Medicine, Gaziantep University, Gaziantep 27310, Turkey; 2Laboratory of Medical Microbiology, Gaziantep Abdülkadir Yüksel State Hospital, Gaziantep 27100, Turkey; mehmeterinmez92@hotmail.com; 3Department of Medical Microbiology, Gaziantep Public Health Laboratory, Gaziantep 27010, Turkey; gonencozer@hotmail.com; 4Department of Medical Biology, Faculty of Medicine, Gaziantep University, Gaziantep 27310, Turkey; shameemsaadat@gantep.edu.tr

**Keywords:** methylene blue, methicillin resistance, *Staphylococcus aureus*, ex vivo

## Abstract

Methylene blue (MB) is a water-soluble dye that has a number of medical applications. Methicillin-resistant *Staphylococcus aureus* (MRSA) was selected as a subject for research due to the numerous serious clinical diseases it might cause and because there is a significant global resistance challenge. Our main goal was to determine and analyze the antibacterial effects of MB against *S. aureus* both in vitro and ex vivo to enhance treatment options. A total of 104 MRSA isolates recovered from various clinical specimens were included in this study. Minimum inhibitory concentration (MIC) values of MB against MRSA isolates were determined by the agar dilution method. One randomly selected MRSA isolate and a methicillin-susceptible *S. aureus* strain (*S. aureus* ATCC 25923) were employed for further evaluation of the antibacterial effects of MB in in vitro and ex vivo time-kill assays. A disc diffusion method-based MB + antibiotic synergy assay was performed to analyze the subinhibitory effects of MB on ten isolates. MICs of MB against 104 MRSA isolates, detected by the agar dilution method, ranged between 16 and 64 µg/mL. MB concentrations of 4 and 16 µg/mL showed a bactericidal effect at 24 h in the ex vivo time-kill assays and in vitro time-kill assays, respectively. We observed a significant synergy between cefoxitin and methylene blue at a concentration of 1–2 μg/mL in two (20%) test isolates. Employing MB, which has well-defined pharmacokinetics, bioavailability, and safety profiles, for the treatment of MRSA infections and nasal decolonization could be a good strategy.

## 1. Introduction

Methylene blue (MB), also known as methylthioninium chloride, is a water-soluble cationic thiazine dye that has been used for a long time in industry and medicine [1,2]. Some countries have approved MB for a number of different indications, including the treatment of both acquired and hereditary methemoglobinemia, the prevention of ifosfamide-induced encephalopathy in the management of human cancer, the prevention of urinary tract infections, the intraoperative visualization of nerves and endocrine glands and of pathologic fistulae, and the sterilization of transfusion blood [2,3,4,5,6]. In a recent study, MB was reported to be effective on *Mycobacterium tuberculosis* complex isolates [7]. According to other reports, MB shows activity against *Plasmodium falciparum* strains and may be employed as an antimalarial agent in combination with other medications [8]. Many studies have reported that MB, on its own or in combination with various other compounds, has antibacterial activity against *Escherichia coli*, *Pseudomonas aeruginosa*, *Staphylococcus aureus*, *Staphylococcus epidermidis*, *Candida albicans,* and *Aspergillus niger* [9,10,11]. It has also been reported that MB might have the potential to be used as a drug for colistin-resistant *Acinetobacter baumannii,* which is considered a superbug [12]. Methylene blue is generally considered safe for in vivo use, with doses of 7.5 mg/kg showing no toxic effects in humans [13]. The drug is well-absorbed from the gastrointestinal tract and is rapidly and widely distributed throughout the body. Maximal plasma concentrations are reached 2 h following oral administration, and the plasma half-life is about 20 h [14,15].

*S. aureus*, a Gram-positive round-shaped bacterium, is a leading cause of bacterial infections throughout the world [16]. Populating commonly in the environment and on our body surface, *S. aureus* can infiltrate the underlying tissues and bloodstream when our cutaneous or mucosal barriers are breached due to wounds, chronic skin conditions, or surgical interventions, leading to infections ranging from mild skin and soft-tissue infections to life-threatening endocarditis, chronic osteomyelitis, pneumonia, or bacteremia [17,18]. Anterior nares and the skin are the main sites of *S. aureus* carriage, which often remains asymptomatic. However, upon penetration of endothelial, epithelial, or dermal barriers, commensal turns into a pathogen. Hence, infections with *S. aureus* often are caused by the respective carriage strains [19]. Despite the advent in the use of antibiotics, *S. aureus* remains a critical threat to human health, being significantly associated with increased hospital costs, prolonged hospital stays, and worsened hospital mortality across the world. Increased frequency in the hospital setting and extended treatment phases is mainly attributed to the global dissemination of multi-drug-resistant (MDR) *S. aureus* strains, the most well-known of which is methicillin-resistant *S. aureus* (MRSA) [20].

Due to delayed proper treatment and less favorable alternative treatment regimens, the mortality rate of MRSA bloodstream infections is double that of equivalent infections attributed to methicillin-susceptible strains [21,22]. In many parts of the world, MRSA infections are widespread in both healthcare facilities and the general population. The primary mechanism of resistance is the synthesis of an accessory penicillin-binding protein, PBP2a/PBP2c, which results in the resistance to all beta-lactams except for the recently developed class of specific “anti-MRSA” cephalosporins. These drugs have an adequate affinity for PBP2a, and most likely also the PBP encoded by mecC, to be efficient against MRSA [23]. The mecA or recently identified mecC genes are responsible for the synthesis of accessory PBPs [21,24]. Being first recognized in the 1960s, MRSA has become a dominant cause of both hospital and community-acquired bacterial infections worldwide in the last few decades [25]. In this study, we aimed to investigate the antibacterial effects of MB against *S. aureus* both in vitro and ex vivo to enhance treatment options.

## 2. Results

### 2.1. Bacterial Isolates

Of the total 104 MRSA strains used in this study, 48 (34%) were collected from the wound site, 35 (12%) from blood, 4 (4%) from surgical materials, 4 (4%) from urine, and 48 (46%) from respiratory tract samples. The distribution of these strains according to the departments was 4 (4%) from the neonatal intensive care unit, 4 (4%) from the oncology service, 27 (26%) from the pediatric service, 20 (19%) from the surgical intensive care unit, 4 (4%) from other intensive care units, and 45 (43%) from other services.

### 2.2. Agar Dilution Method

Minimum inhibitory concentration (MIC) values of MB against 104 MRSA isolates, determined by agar dilution method, ranged between 16 and 64 µg/mL. Among these, 22 isolates had an MIC of 16 µg/mL, 12 isolates had an MIC value of 32 µg/mL, and 70 isolates had an MIC of 64 µg/mL. The median MIC value was 64 µg/mL and MIC90 was 64 µg/mL. MIC of isolate no. 1 and *S. aureus* ATCC 25923 was determined at 16 µg/mL. Data analyses showed significantly higher mean MIC values of MB for gentamicin, ciprofloxacin, levofloxacin, and fosfomycin in susceptible isolates compared with resistant isolates. The *p*-values and mean MIC values were gentamicin (0.000; 62.56), ciprofloxacin (0.006; 62.50), levofloxacin (0.006; 62.50), and fosfomycin (0.006; 62.50), respectively.

#### 2.2.1. In Vitro Time-Kill Method

One randomly selected MRSA isolate and a methicillin-susceptible *S. aureus* strain (*S. aureus* ATCC 25923) were used in the in vitro time-kill assays. The time-kill profiles of isolates showing changes in the number of bacteria (log_10_ CFU/mL) according to MB concentration and time are shown in Figure 1 and Figure 2. MB concentrations of 16 µg/mL and over showed an immediate bactericidal effect without significant regrowth for both strains. MB concentrations of 4 and 8 µg/mL displayed bacteriostatic effects on the MRSA isolate and standard ATCC strain at the end of 24 h, respectively. MB concentrations of 4 µg/mL and lower displayed bacterial killing in the early hours of the experiment, but with subsequent regrowth close to control values across the 24 h (Figure 1 and Figure 2).

#### 2.2.2. Ex Vivo Time-Kill Method

The same isolates used in the in vitro time-kill assays were used in the ex vivo time-kill assays. The time-kill profiles of isolates and changes in log_10_ CFU/mL from the initial inoculum at 24 h are shown in Figure 3 and Figure 4. MB concentrations of 4 µg/mL and over showed a total bactericidal effect at 24 h. MB concentrations of 2 µg/mL displayed bacterial killing in the early hours of the experiment but with subsequent regrowth for *S. aureus* ATCC 25923 across the 24 h (Figure 3 and Figure 4). In vitro and ex vivo methylene blue time-kill results are summarized in Table 1 and Table 2.

#### 2.2.3. Cumulative Analysis of the Mean and Standard Deviation Values in Time-Kill Studies

The mean and standard deviation (SD) values of colony counts (Log CFU/mL) in the beginning (hour zero) and at the end of the time-kill studies (24th h) employing increasing concentrations of MB (2, 4, 8, 16, 32, and 64 μg/mL) were determined as follows, respectively: hour zero: 3.67 ± 1.55, 3.34 ± 2.05, 2.69 ± 1.75, 2.85 ± 2.30, 1.47 ± 1.01, 1.04 ± 0.75; 24th h: 4.48 ± 4.13, 3.16 ± 3.89, 1.23 ± 2.47, 0 ± 0, 0 ± 0, 0 ± 0. Time- and concentration-dependent cumulative analysis of the time-kill studies showing the means and SDs were summarized as a graphic form in Figure 5.

### 2.3. Methylene Blue and Antibiotic Synergy Testing via Disc Diffusion Method

We observed a significant synergy between cefoxitin and methylene blue at a concentration of 1–2 μg/mL in two isolates (isolate no. 31 and no. 99). Weaker synergy was observed in four isolates (isolate nos. 33, 50, 75, and 99). We found a significant synergy between penicillin and methylene blue at only one concentration of 16 μg/mL (isolate no. 31); in the presence of 16 mcg/mL MB concentration, in addition to the enlargement of inhibition zone (≥26 mm: susceptible), the sharp inhibition zone edge around the penicillin disc turned into a fuzzy inhibition zone, which is a sign of susceptibility according to the EUCAST criteria [21]. The increase in the diameter of inhibition zones can be seen in Figure 6. The synergy test results are shown in Table 3.

## 3. Discussion

MRSA infections are challenging to treat. The most notable aspect is that MRSA strains have developed concurrent resistance to a wide range of routinely used antibiotics, including fluoroquinolones, aminoglycosides, chloramphenicol, macrolides, and tetracycline [26]. The Europe/ European Economic Area (EU/EEA) countries reported 81163 isolates of *S. aureus*. EU/EEA population-weighted mean MRSA percentage was 15.8% in this report. Despite the relative decrease in the percentage of MRSA infections, MRSA remains an important pathogen in Europe, and the percentages are still high in several countries. *S. aureus* is a prevalent cause of infections of the bloodstream, with a substantial morbidity and mortality rate. The exact way of action of MB against bacteria is still being discussed. According to research, its way of action is based on its distinctive redox characteristics, which interfere with a variety of electron transport channels in bacteria [27,28,29]. Furthermore, phenothiazines (MB analogues) are thought to carry out antibacterial activity via either attaching to or intervening with the penicillin-binding protein (PBP) [30].

Gazel et al. reported that *A. baumanii* complex clones that had developed colistin resistance by exposure to colistin had acquired susceptibility to MB. They found that MB selectively inhibited the growth of colistin-resistant subpopulations of the *A. baumanii* complex, while colistin susceptible clones went on growing [12]. However, in our study, MB inhibited the growth of both methicillin-resistant and susceptible strains. MB alone inhibited the growth of *Mycobacterium smegmatis* at a 15.62 µg/mL concentration in a bacteriostatic way corresponding to its fungistatic feature, according to a study on *C. albicans* and *M. smegmatis* by Pal et al. [31]. They also discovered that MB caused defective cell surface phenotypes, abnormal colony morphologies, and DNA damage in *Mycobacteria*. They encouraged more research on the *M. tuberculosis* complex, since this pathogen has unique cell envelope components with complex lipids that provide pathogenicity [31]. Gazel et al. investigated the in vitro activity of methylene blue on *M. tuberculosis* complex clinical isolates from a university hospital. They reported that the critical concentration of MB at 2 µg/mL inhibited 35% of the clinical isolates and MB could have the potential as an alternative antituberculosis drug [7].

Almost all of the reports evaluating the antibacterial activity of MB against MRSA focus on the synergistic effects of MB in combination with the application of a laser (photodynamic therapy) [32]. However, in our study, we planned to investigate the antibacterial effect of MB alone, as a novel antimicrobial compound. Ronqui et al. investigated the synergistic antimicrobial effect of photodynamic therapy mediated by MB on the methicillin-susceptible *S. aureus* strain (ATCC 25923). The most favorable outcomes were observed following a combination treatment of photodynamic therapy accompanied by ciprofloxacin on biofilms, which enhanced the reduction of bacteria on biofilms, concluding in a 5.4 log reduction for *S. aureus* biofilm. However, they did not investigate the bactericidal effect of MB alone in their study [33]. In a different study, the co-administration of amoxicillin and light-activated MB increased each other’s uptake in *S. aureus*, resulting in up to 8 log reductions in MRSA infections. This approach was proven to be efficient against *S. aureus* infections independent of their antibiotic resistance profiles and did not result in noteworthy bacterial resistance after five days of continuous treatment.

Although MB has been shown to have antimicrobial properties, most research has not identified its MIC and has used other approaches instead. The MIC is defined as the lowest concentration of an antimicrobial drug that would visibly inhibit an organism’s growth following overnight incubation and is the “gold standard method” for the determination of susceptibility [32,34]. Khan et al. investigated the minimum inhibitory concentration and minimum fungicidal concentration (MFC) of MB on pathogenic *C. albicans* biofilm. MIC and MFC of MB were determined as 62.5 µg/mL and 125.0 µg/mL, respectively. Several studies have determined the MIC of MB against *S. aureus*. The first study presenting MIC data for a *S. aureus* strain was performed by Shatti and Authman. They tested the *S. aureus* strain at only three different concentrations of MB (which is not a recommended method) and determined the MIC value as 10 mg/mL (equal to 10000 µg/mL in the standard method) [35]. Comparing these findings with our study, the activity of MB was determined to be much stronger in our study (MIC90: 64 µg/mL). Although this was the first report determining MIC for MB, it seems that there are some problems regarding the use of units (mg/mL) in the article text. When we looked into the available literature, we noted that another study had reported similar results to our study; Thesnaar et al. had found the MIC value of *S. aureus* isolate to be 16 µg/mL using a broth microdilution method [32]. In addition to revealing the antibacterial effects of MB by defining MIC values, our study is the first to present both in vitro and ex vivo MRSA time-kill results, which could be very helpful for guiding correct dosing strategies for MB. To our knowledge, our research is one of the first reports to provide MIC values of MB against MRSA isolates using the well-established agar dilution method. Significantly higher mean MIC values to MB were determined in susceptible isolates compared with resistant isolates for gentamicin, ciprofloxacin, levofloxacin, and fosfomycin antibiotics in our study. We speculate that mutations and micro-physiological alterations causing resistance to gentamicin, ciprofloxacin, levofloxacin, and fosfomycin may result in a higher level of susceptibility to MB and using MB as antimicrobial agent on these types of isolates could be more beneficial for the treatment of MRSA infections in the future.

There are a few studies evaluating the plasma/serum levels of MB on healthy volunteers. According to these studies, higher concentrations of MB could be obtained in blood when MB is administered orally [36]. Moreover, animal experiments demonstrated that a twenty-fold higher concentration can be obtained in the brain after i.v. administration [36]. In a recent study, Di Stefano et al. reported that methylene blue AUC0-t was 10.7 ± 6.7 μg/mL × h after 100 mg and 25.2 ± 7.4 μg/mL × h after 200 mg oral administration of methylene blue in tablet form. Only one adverse event was detected related to methylene blue—a mild increase in alanine aminotransferase [37]. In another study, they investigated the safety and bioavailability of methylene blue after single oral doses of 200 and 400 mg in healthy volunteers and reported that MB concentrations of 32.94 μg/mL and 38.08 μg/mL were obtained after administration of MB, respectively. Only non-serious adverse events occurred during the study [38]. When these studies are evaluated together, we can conclude that MB may have the potential to be used as an antimicrobial drug, since 16 and 32 μg/mL MICs were detected in our study.

In the next step, we performed a synergy test to investigate the subinhibitory effects of MB on resistance profiles of beta-lactam antibiotics. According to EUCAST guidelines, most *S. aureus* are penicillinase producers and some are methicillin-resistant. Either mechanism renders them resistant to benzylpenicillin, phenoxymethylpenicillin, ampicillin, amoxicillin, piperacillin, and ticarcillin. Isolates that test susceptible to benzylpenicillin and cefoxitin can be reported susceptible to all penicillins. Isolates that test resistant to benzylpenicillin but susceptible to cefoxitin are susceptible to beta-lactam/beta-lactamase inhibitor combinations, the isoxazolylpenicillins (oxacillin, cloxacillin, dicloxacillin, and flucloxacillin), and nafcillin. Isolates that test resistant to cefoxitin are resistant to all penicillins. [21]. In this test, we observed a significant synergy between cefoxitin and methylene blue at a concentration of 1–2 μg/mL in two (20%) isolates (development of susceptibility), while a weaker synergy was observed in four (40%) isolates. We evaluated that MB has the potential to be used at lower/subinhibitory doses as a combination compound for sensitization of MRSA bacteria to beta-lactam antibiotics, since isolates that test resistant to benzylpenicillin but are susceptible to cefoxitin are also susceptible to beta-lactam/beta-lactamase inhibitor combinations and isoxazolylpenicillins [21]. Regarding one isolate (isolate no. 31), in the presence of a 16 mcg/mL MB concentration, in addition to the enlargement of inhibition zone (≥26 mm: susceptible), the sharp inhibition zone edge around the penicillin disc turned into a fuzzy inhibition zone, which is a sign of susceptibility according to the EUCAST criteria [21]. So, these data verified our finding regarding the susceptibility development of this isolate to benzylpenicillin and other penicillin analogues.

Other than systemic administration, MB can also be used in topical form. There are studies reporting the antimicrobial effects of MB at a concentration of 500 μg/mL in topical form for treatment of oral candidiasis [39]. Furthermore, MB collutory preparations at a concentration of 10,000 μg/mL have been licensed in Turkey for treatment of oral candidiasis [40]. So, higher concentrations of MB (well above the MICs that we detected) can be achieved in the targeted superficial tissues and these collutory, gel, or cream forms of MB can be used to treat mucosal and skin infections of MRSA without any systemic side effect [41]. Currently, a mupirocin antibiotic is topically used for nasal decolonization of *S. aureus*; however, we claim that MB can be an alternative to this drug, since much higher concentrations of MB can be applied on nasal mucosa in topical form [42].

As another claim, MB could be useful for the treatment of severe neurological infections such as MRSA meningitis. A study on the pharmacokinetics and organ distributions of intravenous and oral MB revealed that, following intravenous MB delivery, brain concentrations were found to be twenty-fold greater than in blood in the animal models [36]. Even if cases of meningitis caused by MRSA are relatively rare, a case of bacterial meningitis is an emergency medical condition with a significant mortality rate if not treated. Since MB can cross the blood–brain barrier, MB derivatives could be an alternative antimicrobial agent for the treatment of MRSA meningitis in the future.

Comparing our results with other studies, we reached different findings regarding the inhibitory concentrations of MB. Interestingly, we determined higher inhibitory concentrations using in vitro models (agar dilution and time-kill assay) but lower inhibitory concentrations using an ex vivo time-kill assay. We observed that concentrations of MB at 4 µg/mL (ex vivo) showed a bactericidal effect even 24 h after the application of the compound. Our in vitro and ex vivo results were concordant in terms of the rapid start of bactericidal activity; however, in the long-term observation, there was no regrowth or a weak regrowth in the ex vivo model, suggesting that MB could be more potent if it can be used as a drug in vivo. The previous studies were conducted only under in vitro conditions and they may not reflect what can happen in the cellular/biological ecosystem. Ex vivo tests can partially mimic in vivo tissue environment and they are superior to the in vitro tests since the additional effects of the human cells on bacteria can be observed in combination with the antimicrobial drug. It has been suggested that ex vivo models may be more clinically relevant, since in vitro studies do not take into account host proteins that can neutralize antiseptic/antimicrobial activity [43,44]. In the recent past, it has been revealed that *S. aureus*, which was considered to be an extracellular pathogen, can invade and survive inside mammalian cells. These infected cells may provide a reservoir and protective niche for *S. aureus*, and, thus, the bacterial cells may be relatively protected from conventional antibiotics [45]. This recent study and some other studies used and recommended ex vivo assays in order to observe the effect of antibiotics on the MRSA bacteria, which can be protected inside human cells [45,46]. That is why we employed an ex vivo time-kill assay in our study.

MRSA infections are one of the most common causes of nosocomial infections. Methicillin-resistant *Staphylococcus aureus* can survive on some surfaces like patients’ beds, tables, desks, razors, other furniture, and athletic equipment for hours, days, or even weeks. It can spread to people who touch a contaminated surface, and MRSA can cause infections if it gets into a cut, scrape, or open wound [47]. Due to the bactericidal effect that we reported in our study, MB can be used as an alternative antiseptic and disinfectant agent in order to prevent the spreading of MRSA in hospital settings. Lower doses of MB may have the potential to be used as an adjuvant agent in combination with beta-lactam antibiotics. Higher doses of MB (≥ 64 µg/mL) can be used for treatment of superficial MRSA infections, including skin and wound infections, and for decolonization of the nasal mucosa of MRSA carriers in topical form. In our study, the majority of the isolates were inhibited by MB at a concentration of 64 µg/mL (MIC90 = 64 µg/mL) in vitro. Moreover, in vitro time-kill studies indicated rapid bactericidal effect (3 log decrease in the number of bacteria in a few seconds, T = 0) of MB at doses of 32 and 64 µg/mL. So, MB at a concentration of 64 µg/mL may have the potential to be used as an antiseptic or disinfectant for eradication of MRSA in hospital settings. Additionally, this dose of MB can be used for treatment of superficial MRSA infections, including folliculitis, furuncle, carbuncle, impetigo, and wound infections and in the decolonization of the nasal mucosa in MRSA carriers in topical form. MB at concentrations as low as 1 and 2 µg/mL showed a synergistic effect on test isolates in combination with the cefoxitin antibiotic. Such low concentrations of MB (1–2 µg/mL) can be appropriate for systemic use and can be used in combination with some other beta-lactam antibiotics. Just like the activity of the beta-lactam/beta-lactamase inhibitor group of antibiotics (ampicillin/sulbactam or amoxicillin/clavulanate), MB + cefoxitin or MB + flucloxacillin preparations can be developed in a similar form and could be used in the treatment of systemic MRSA infections.

Limitations of our study are that all the clinical isolates used in our study were from the same university hospital in our region and we could not perform genotyping of the study isolates. Another limitation is that we could not employ an in vivo model to evaluate the antimicrobial effects of MB in a real biological ecosystem. Even though there were no defined standard protocols for the ex vivo investigation of antibacterial effects, we tried to create a bridge between in vitro and in vivo assays. Also, the correlation between ex vivo and in vivo models was high, suggesting a high potential value for scientific research [48]. As another limitation, methylene blue compounds are safe in lower doses [49] and more toxicity studies should be performed before recommending this antimicrobial for treatment of MRSA infections.

## 4. Materials and Methods

### 4.1. Bacterial Isolates

This research was conducted at a tertiary care university hospital in Gaziantep, Turkey. A total of 104 MRSA isolates from the University Hospital Medical Microbiology Laboratory bacterial stock culture archives were included in this study. These isolates were recovered from various clinical specimens and obtained between December 2017 and January 2021. Isolates were stored at −80 °C using a bead tube containing 150 mL/L pure glycerol and 850 mL/L nutrient broth (GBL, Turkey). In addition, methicillin-susceptible *S. aureus* ATCC 25923 was employed as a methicillin-sensitive standard strain. Prior to testing, frozen isolates were subcultured twice on 5% sheep blood agar plates to ensure purity and viability. Antimicrobial susceptibility profiles of all study isolates (for gentamicin, ciprofloxacin, levofloxacin, erythromycin, clindamycin, tetracycline, fosfomycin, and fusidic acid) were screened in the laboratory archive records.

### 4.2. Determination of Minimum Inhibitory Concentrations

MIC values of MB against MRSA isolates were determined by the agar dilution method according to CLSI recommendations [50]. MB (Kimyalab, Istanbul, Turkey) stock solutions were prepared immediately before usage in sterile distilled water on each study day. Briefly, Mueller–Hinton agar (MHA) (BD Difco, Sparks, MD, USA) plates were prepared with two-fold dilutions of MB ranging from 0.125 to 512 µg/mL. Inocula were prepared by suspending colonies grown from overnight cultures in saline to a 0.5 McFarland standard (1.5 × 10^8^ CFU/mL). Final inocula were adjusted to contain 10^4^ bacteria/spot. Plates were incubated at 35 °C for 18 h and the lowest concentration of MB inhibiting the growth of bacteria was defined as the MIC value (Figure 7).

#### 4.2.1. In Vitro Time-Kill Assay

One randomly selected MRSA isolate and a methicillin-susceptible *S. aureus* strain (*S. aureus* ATCC 25923) were employed for further evaluation of the time-dependent antibacterial effects of MB in the time-kill assay. For the assay, bacterial suspensions were prepared in 20 mL Mueller–Hinton broth (MHB) from the fresh colonies grown on sheep blood agar plates, diluted, and grown to log phase by incubation at 37 °C for two hours. The turbidity of each bacterial culture tube was adjusted to a final inoculum with a density of 5 × 10^5^ CFU/mL. MB was added to the broth cultures to yield concentrations of 2, 4, 8, 16, 32, and 64 µg/mL. The test and control tubes were incubated at 35 °C.

Viable counting was performed on samples collected at 0, 1, 3, 6, 12, and 24 h after MB addition. Samples of bacterial cell suspension (50 μL) were plated on MHA plates. Colonies were counted manually after incubation of subcultures for 24 h at 35 °C. Colony-forming units (CFU/mL) were determined from the average count of the duplicate plates, followed by the calculation of the log_10_ CFU/mL. Bactericidal activity was evaluated as a ≥10^3^ (3 log_10_) decrease in CFU/mL over the time period examined [51]. Time-kill studies were repeated on three independent occasions.

#### 4.2.2. Ex Vivo Time-Kill Assay

The same two strains used in the in vitro time-kill assay (one randomly selected MRSA isolate and a standard methicillin-susceptible *S. aureus* ATCC 25923 strain) were employed for further evaluation of the time-dependent antibacterial effects of MB in a cell culture environment using a time-kill assay. MB was tested at concentrations of 2–64 µg/mL. For the cell culture experiments, U-2 OS (ATCC HTB-96) cells were obtained from the American Type Culture Collection (ATCC) (Manassas, VA, USA). The cells were maintained in Dulbecco’s modified Eagle’s medium (Thermo Scientific, Sunnyvale, California, USA), supplemented with 10% fetal bovine serum (Invitrogen, Rockville, MD, USA). The cells were incubated at 37 °C in a humidified atmosphere containing 5% CO_2_. No antimicrobial agents were used during this procedure. Bacterial suspensions were prepared in MHB from fresh colonies grown on sheep blood agar plates, diluted, and grown to the log phase. The turbidity of bacterial cultures was adjusted to a final inoculum with a density of 5 × 10^5^ CFU/mL inside the 5 mL osteosarcoma cell line culture. MB was added to the liquid cell line cultures to yield concentrations of 2, 4, 8, 16, 32, and 64 µg/mL. The test and control tubes were incubated at 35 °C.

Viable counting was performed on samples collected at 0, 1, 3, 6, 12, and 24 h after MB addition. Samples of bacterial cell suspension (50 μL) were plated on MHA plates. Colonies were manually counted after incubation of subcultures for 24 h at 35 °C. CFU was determined from the average count of the duplicate plates, followed by the calculation of the log_10_ CFU/mL. Bactericidal activity was evaluated as a ≥3 log decrease in CFU/mL over the time period examined [51]. Time-kill studies were repeated on three independent occasions.

### 4.3. Methylene Blue and Antibiotic Synergy Testing via Disc Diffusion Method

A total of 10 MRSA clinical isolates from different wards having different antimicrobial susceptibility profiles were included in this study. Cefoxitin (30 µg) and benzylpenicillin (1 unit) (Bioanalyse, Ankara, Turkey) antibiotic discs were used for synergy testing. The Kirby–Bauer disk diffusion method was used to analyze the development of susceptibility to the antibiotics in combination with increasing concentrations of MB. First, Mueller–Hinton agar (MHA) plates containing MB at concentrations of 0, 1, 2, 4, 8, 16, 32, and 64 μg/mL were prepared. Bacterial suspensions of 0.5 McFarland standard were prepared and inoculated on MHA media. Within 15 min after inoculation, antibiotic discs were placed on the surface of the MHA and the media were incubated at 35 °C for 18 h. After incubation, inhibition zones had formed around the antibiotic discs and the diameters of the inhibition zones were measured as millimeters and recorded. The interpretation of inhibition zone diameters (susceptible or resistant) was made in compliance with EUCAST criteria [21]. Regarding cefoxitin, an inhibition zone diameter ≥ 26 mm was interpreted as susceptible, while a zone diameter < 26 mm was interpreted as resistant. Regarding benzylpenicillin, an inhibition zone diameter ≥ 22 mm was interpreted as susceptible, while a zone diameter < 22 mm was interpreted as resistant. According to EUCAST criteria, isolates that tested susceptible to benzylpenicillin and cefoxitin were reported susceptible to all penicillins. Isolates that tested resistant to benzylpenicillin but susceptible to cefoxitin were reported susceptible to beta-lactam/beta-lactamase inhibitor combinations, the isoxazolyl penicillins (oxacillin, cloxacillin, dicloxacillin, and flucloxacillin), and nafcillin. Isolates that tested resistant to cefoxitin were determined to be resistant to all penicillins [21].

### 4.4. Statistical Analysis

A power analysis was performed to estimate the smallest sample size. The minimum number of samples that needed to be included in this study to estimate a prevalence of 6% with 5% accuracy and 95% confidence interval was calculated as 88 MRSA isolates (G-power program version 3.1. was used). The comparison of methylene blue MIC values between resistant and susceptible isolates to several antibiotics (gentamicin, ciprofloxacin, levofloxacin, erythromycin, clindamycin, tetracycline, fosfomycin, and fusidic acid) was performed using an independent sample *t*-test. Data analysis was performed via SPSS (Statistical Packages of Social Sciences) version 22.0. A *p*-value of <0.05 was considered statistically significant. Statistical analyses determining the mean and SD values of the time-kill and disc diffusion synergy studies were performed by using GraphPad Prism Software V8.0.2 (GraphPad, San Diego, CA, USA).

## 5. Conclusions

In our study, MB concentrations of 16 µg/mL and over showed a bactericidal effect at 24 h in the in vitro and ex vivo time-kill assays. The bactericidal effect of the MB was more evident and stronger in the ex vivo time-kill assay, suggesting that the antimicrobial effect of MB could be stronger in a real in vivo MRSA infection. Subinhibitory doses of MB increased the sensitivity of MRSA bacteria to beta-lactam antibiotics in our study, suggesting that MB can also be used as a combination compound for treatment of resistant MRSA infections. Since higher topical doses of MB (500–1000 µg/mL) can be used on skin and mucosa, MB can be more effective in the treatment of superficial MRSA infections via local application. In the light of our findings, in vivo experiments are needed to demonstrate the antibacterial effects of MB on MRSA in a living organism in order to determine the side effects and optimal dosing regimens. Regarding treatment of systemic MRSA infections, studies investigating the effects of MB + beta-lactam combinations could be more beneficial, since these combinations may be an alternative to classical beta-lactam/beta-lactamase inhibitor combination antibiotics in the future. As another point, in vivo researches could be planned to investigate the topical activity of MB in the decolonization of MRSA in nasal carriers. As a result, there consequently remains a strong need for new antibiotics, particularly those directed against multi-drug-resistant *S. aureus,* which is a common problem in hospital settings. We conclude that MB could be a promising antimicrobial agent for the treatment of MRSA or other drug-resistant bacterial infections in the future.

## Figures and Tables

**Figure 1 pharmaceuticals-17-00241-f001:**
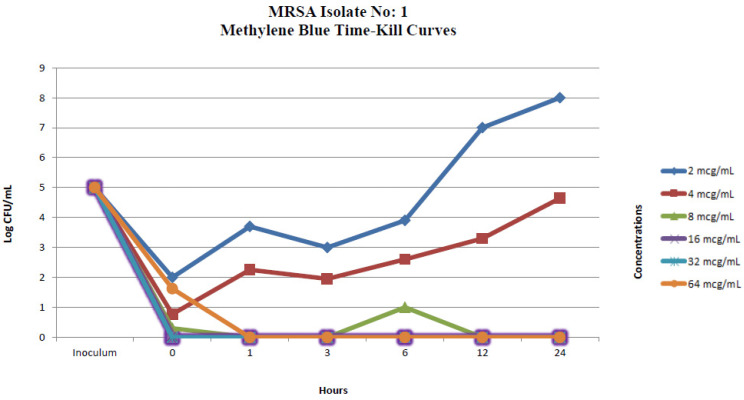
In vitro methylene blue time-kill curves of MRSA isolate no. 1.

**Figure 2 pharmaceuticals-17-00241-f002:**
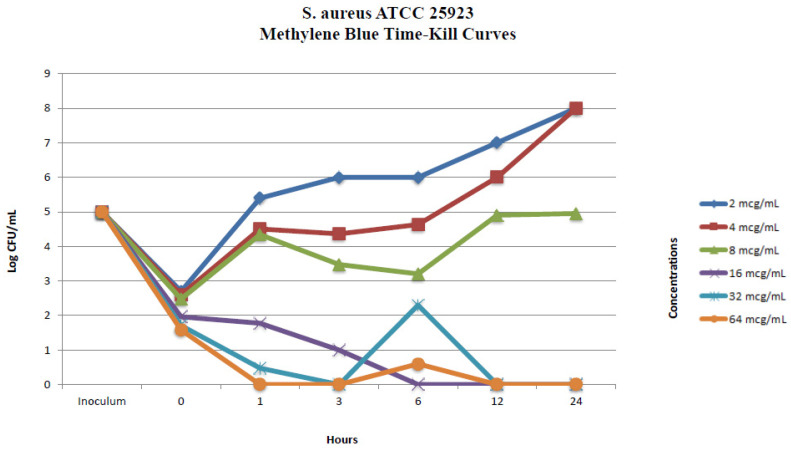
In vitro methylene blue time-kill curves of *S. aureus* ATCC 25923.

**Figure 3 pharmaceuticals-17-00241-f003:**
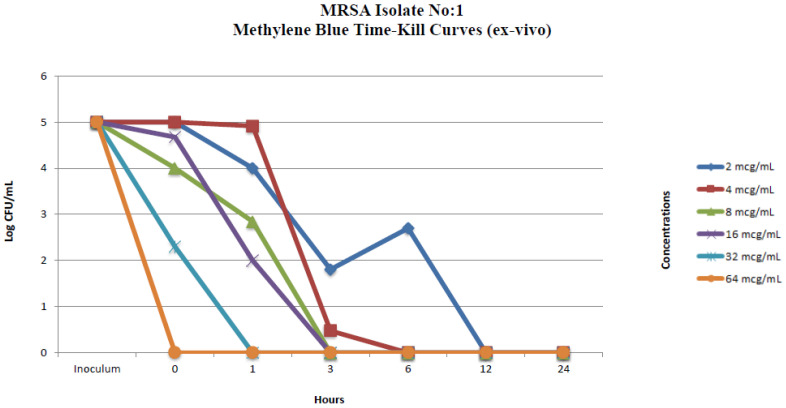
Ex vivo methylene blue time-kill curves of MRSA isolate no. 1.

**Figure 4 pharmaceuticals-17-00241-f004:**
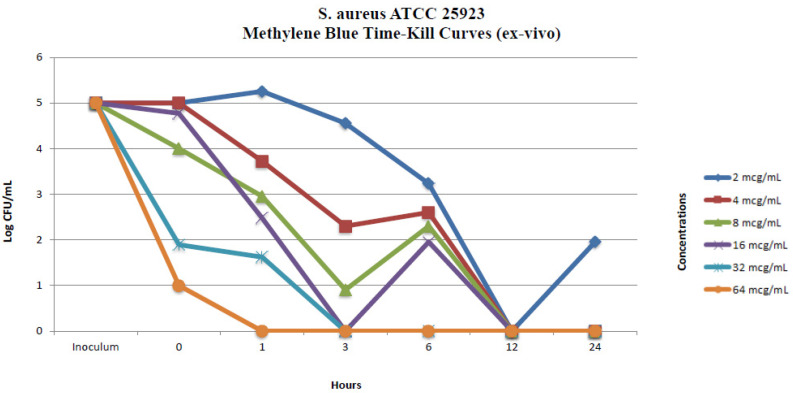
Ex vivo methylene blue time-kill curves of *S. aureus* ATCC 25923.

**Figure 5 pharmaceuticals-17-00241-f005:**
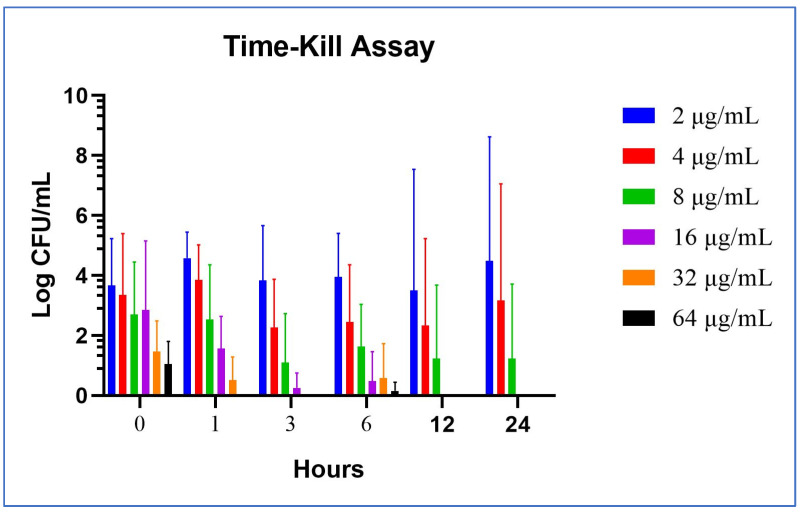
Cumulative descriptive analysis of time-kill studies. Time (0–24 h) and concentration (2–64 μg/mL MB)-dependent mean and SD values of the CFUs are shown as bars and lines in the graphic image.

**Figure 6 pharmaceuticals-17-00241-f006:**
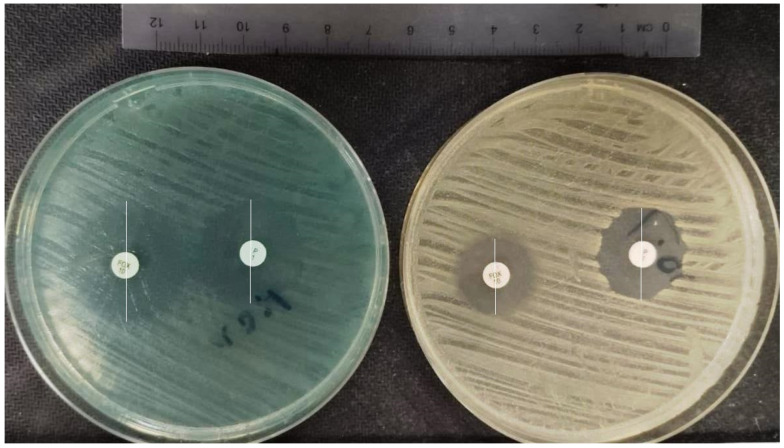
Inhibition zone diameters around the antibiotics in the presence and absence of MB. P—benzylpenicillin; FOX—cefoxitin. The inhibition zone diameters to benzylpenicillin and cefoxitin are determined in millimetres. The MHA plate on the left side contains 2 μg/mL MB, while there is no MB inside the MHA plate on the right side. The inhibition zone diameters around the cefoxitin and penicillin disc are significantly increased by the presence of 2 μg/mL methylene blue (**left**).

**Figure 7 pharmaceuticals-17-00241-f007:**
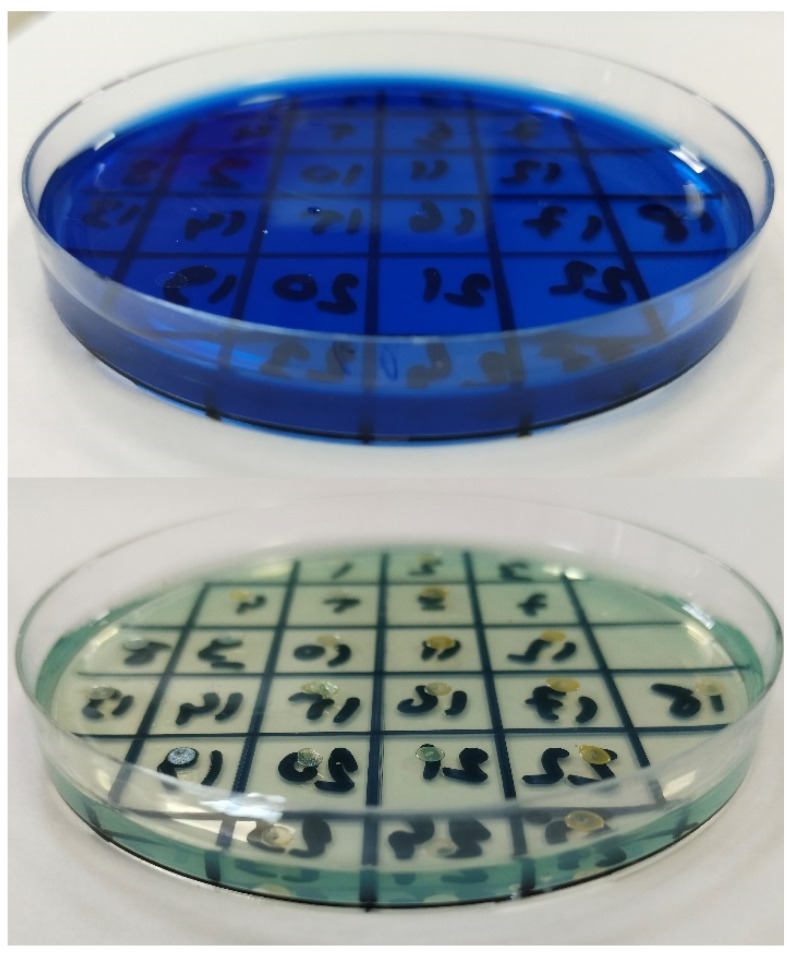
Agar dilution test plates of MRSA isolates. Images of MRSA-inoculated agar plates containing high (64 µg/mL) concentration of MB (**top**) and low (1 µg/mL) concentration of MB (**bottom**). The visible colonies (white spots) growing on the agar plate indicate resistance to MB. All MRSA isolates are resistant to 1 µg/mL concentration of MB (**bottom**), while they are susceptible (no growth) to the higher concentration (64 µg/mL) of MB (**top**).

**Table 1 pharmaceuticals-17-00241-t001:** In vitro and ex vivo methylene blue time-kill results of MRSA isolate no. 1.

	MRSA Isolate No. 1 Colony Counts (Log CFU/mL) *											
	Methylene Blue Concentrations											
	2 μg/mL		4 μg/mL		8 μg/mL		16 μg/mL		32 μg/mL		64 μg/mL	
Time (h)	in vitro	ex vivo	in vitro	ex vivo	in vitro	ex vivo	in vitro	ex vivo	in vitro	ex vivo	in vitro	ex vivo
0	2	5	0.77	5	0.30	4	0	4.67	0	2.30	1.62	0
1	3.69	4	2.25	4.91	0	2.84	0	2	0	0	0	0
3	3	1.80	1.95	0.47	0	0	0	0	0	0	0	0
6	3.90	2.69	2.60	0	1	0	0	0	0	0	0	0
12	7	0	3.30	0	0	0	0	0	0	0	0	0
24	8	0	4.64	0	0	0	0	0	0	0	0	0

* The initial inoculum amount for all experiments was 5 Log CFU/mL.

**Table 2 pharmaceuticals-17-00241-t002:** In vitro and ex vivo methylene blue time-kill results of *S. aureus* ATCC 25923.

	*S. aureus* ATCC 25923 Colony Counts (Log CFU/mL) *											
	Methylene Blue Concentrations											
	2 μg/mL		4 μg/mL		8 μg/mL		16 μg/mL		32 μg/mL		64 μg/mL	
Time (h)	in vitro	ex vivo	in vitro	ex vivo	in vitro	ex vivo	in vitro	ex vivo	in vitro	ex vivo	in vitro	ex vivo
0	2.69	5	2.60	5	2.47	4	1.96	4.77	1.70	1.89	1.57	1
1	5.39	5.25	4.50	3.72	4.34	2.95	1.77	2.47	0.47	1.62	0	0
3	6	4.55	4.36	2.30	3.47	0.90	1	0	0	0	0	0
6	6	3.23	4.63	2.60	3.20	2.30	0	1.95	2.30	0	0.60	0
12	7	0	6	0	4.90	0	0	0	0	0	0	0
24	8	1.95	8	0	4.95	0	0	0	0	0	0	0

* The initial inoculum amount for all experiments was 5 Log CFU/mL.

**Table 3 pharmaceuticals-17-00241-t003:** Methylene blue and antibiotic synergy test results with disc diffusion method.

Isolate No	MB [C]	0 µg/mL	1 µg/mL	2 µg/mL	4 µg/mL	8 µg/mL	16 µg/mL	32 µg/mL	64 µg/mL
31		P: 21 FOX: 19	P: 21 **FOX: 22 ^a^**	P: 24 **FOX: 26**	P: 24 **FOX: 26**	P: 25 **FOX: 26**	**P: 26 ^a^** **FOX: 26**	**P: 27** **FOX: 26**	**NO GROWTH ^c^**
33		P: 10 FOX: 10	P: 10 FOX: 10	P: 10 FOX: 11	P: 11 FOX: 15	P: 11 FOX: 15	P: 12 FOX: 16	P: 14 FOX: 17	**NO GROWTH ^c^**
37		P: 7 FOX: 7	P: 7 FOX: 7	P: 7 FOX: 7	P: 7 FOX: 7	P: 7 FOX: 7	P: 7 FOX: 7	P: 7 FOX: 7	**NO GROWTH ^c^**
47		P: 7 FOX: 7	P: 7 FOX: 7	P: 7 FOX: 7	P: 7 FOX: 7	P: 7 FOX: 7	P: 10 FOX: 7	P: 12 FOX: 7	**NO GROWTH ^c^**
50		P: 7 FOX: 7	P: 7 FOX: 7	P: 7 FOX: 7	P: 7 FOX: 7	P: 7 FOX: 7	P: 10 FOX: 7	P: 10 FOX: 10	**NO GROWTH ^c^**
75		P: 10 FOX: 10	P: 10 FOX: 10	P: 10 FOX: 10	P: 11 FOX: 11	P: 12 FOX: 12	P: 12 FOX: 12	P: 16 FOX: 16	**NO GROWTH ^c^**
80		P: 7 FOX: 7	P: 7 FOX: 7	P: 7 FOX: 7	P: 7 FOX: 7	P: 7 FOX: 7	P: 7 FOX: 7	**NO GROWTH ^b^**	**NO GROWTH**
89		P: 7 FOX: 7	P: 7 FOX: 7	P: 7 FOX: 7	P: 7 FOX: 7	P: 7 FOX: 7	P: 9 FOX: 7	**NO GROWTH ^b^**	**NO GROWTH**
93		P: 7 FOX: 7	P: 7 FOX: 7	P: 7 FOX: 7	P: 7 FOX: 7	P: 7 FOX: 7	P: 9 FOX: 7	**NO GROWTH ^b^**	**NO GROWTH**
99		P: 12 FOX: 16	P: 13 FOX: 16	P: 13 **FOX: 27 ^a^**	P: 14 **FOX: 27**	P: 15 **FOX: 27**	P: 15 **FOX: 27**	P: 19 **FOX: 28**	**NO GROWTH ^c^**
**MEAN ± SD, P:** **MEAN ± SD, FOX:**		9.5 ± 4.42 9.7 ± 4.35	9.6 ± 4.50 10 ± 5.09	9.9 ± 5.36 11.6 ± 7.98	10.2 ± 5.45 12.2 ± 8.02	10.5 ± 5.83 12.2 ± 8.02	11.7 ± 5.57 12.3 ± 8.06	15 ± 6.58 15.8 ± 8.59	

^a^ The breakpoint of the inhibition zone diameters indicating the development of susceptibility, **^b^** MIC of MB alone is 32 µg/mL, **^c^** MIC of MB alone is 64 µg/mL. MB—methylene blue; P—benzylpenicillin; FOX—cefoxitin; SD—standard deviation. The inhibition zone diameters to benzylpenicillin and cefoxitin are given in millimeters.

## Data Availability

The data presented in this study are available on request from the corresponding author. Some data of this study were presented as an oral presentation at the “6th Turkish National Clinical Microbiology Congress” on 21 October 2021.

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
