# Peer review of "In Vitro and Ex Vivo Investigation of the Antibacterial Effects of Methylene Blue against Methicillin-Resistant Staphylococcus aureus"

_pharmaceuticals, 2024, doi:10.3390/ph17020241_

Round 1
Reviewer 1 Report
Comments and Suggestions for Authors
In this manuscript, authors measured MICs of MB to MRSA clinical isolates and a MSSA strain. The finding, determining of MIC is significant, however the manuscript contains nebulous portions in study design and methods. Authors should consider whether these comments can be addressed.
1. Authors measured MIC by using agar dilution method. This is no problem, however, why broth microdilution method was not used, though MB is water-soluble?
2. Why did authors use ex-vivo test to measure MIC? What is the meaning to determine ex-vivo test? Objective to use ex-vivo test should be written, with references.
3. Why did author measured MIC mostly for MRSA clinical isolates and only one MSSA? Ideally, more MSSA isolates should be added. Authors wrote that MRSA is highly infectious, but it is completely wrong. Infectivity of MRSA and MSSA are same. Difference is the presence of mecA on SCCmec to confer beta-lactam resistance, which is not related to activity of MB.
4. Clinical isolates of MRSA are heterologous population. Some of them must have any drug-efflux pump that is effective also to MB, that may be the reason why MIC values had a wide range. Did author determine any genotype of MRSA? Did author find any correlation with higher or lower MIC to any attributes of isolates (genotype, specimen type, antimicrobial susceptibility, etc.)?
5. Figure 5, readers cannot see growth of isolates in the left part. Authors should take photo from the side of agar surface, so that readers can see better the growth of bacteria even background is dark.
Author Response
Dear Editor and Reviewers (Please see the attachment)
Appended to this letter is my point-by-point response to the comments raised by the reviewers. As you notice, I agreed with most of the comments raised by the reviewers. The revisions were performed in accordance to the recommendations of Reviewer 1 (marked with yellow), Reviewer 2 (marked with green) and Reviewer 3 (marked with turquoise color) in the text. The revisions recommended by multiple reviewers were marked with gray and newly added sections by own decision were marked pink color.
I would like to take this opportunity to express my sincere thanks to the reviewers who identified areas of our manuscript that needed corrections or modification. I would like to thank you for allowing me to modify and submit the manuscript as an original article.
I am looking forward to see my article being published,
Sincerely Yours,
RESPONSES
In this manuscript, authors measured MICs of MB to MRSA clinical isolates and a MSSA strain. The finding, determining of MIC is significant, however the manuscript contains nebulous portions in study design and methods. Authors should consider whether these comments can be addressed.
1. Authors measured MIC by using agar dilution method. This is no problem, however, why broth microdilution method was not used, though MB is water-soluble?
***Since MB is a colored substance, it can be difficult to determine turbidity by broth microdilution test. Especially, when there is a borderline turbidity, the dark color of MB can mask the blurry appearance and may cause false results. But, in agar dilution test we can visualize 3D appearances of the colonies growing on agar surface (Actually, in the first step of the agar dilution test, we prepared MB solutions in distilled water first). Secondly, while microdilution test is gold standard for most antibiotics, there is no recommended gold standard MIC determining method for testing antimicrobial effects of MB.
2. Why did authors use ex-vivo test to measure MIC? What is the meaning to determine ex-vivo test? Objective to use ex-vivo test should be written, with references.
*** MIC was determined only by performing agar dilution test. Ex vivo test was used to determine time dependent killing, rather than determining the MIC. Ex-vivo tests can partially mimicry in vivo tissue environment and they are superior to the in vitro tests since the additional effects of the human cells on bacteria can be observed in combination with antimicrobial drug. It has been suggested that ex vivo models may be more clinically relevant, since in vitro studies do not take into account host proteins that can neutralize antiseptic/antimicrobial activity (Lepelletier D, Maillard JY, Pozzetto B, Simon A. Povidone Iodine: Properties, Mechanisms of Action, and Role in Infection Control and Staphylococcus aureus Decolonization. Antimicrob Agents Chemother. 2020 Aug 20;64(9):e00682-20. doi: 10.1128/AAC.00682-20) (Ex vivo porcine vaginal mucosal model of infection for determining effectiveness and toxicity of antiseptics. J Appl Microbiol 115:679–688. doi: 10.1111/jam.12277).
Secondly, in the recent past, it has been revealed that S. aureus, which was considered to be an extracellular pathogen, can invade and survive inside mammalian cells. These infected cells may provide a reservoir and protective niche for S. aureus, and thus, the bacterial cells may be relatively protected from conventional antibiotics. (Synthesis of novel monocarbonyl curcuminoids, evaluation of their efficacy against MRSA, including ex vivo infection model and their mechanistic studies. Gagandeep, Prince Kumar, Shamseer Kulangara Kandi, Kasturi Mukhopadhyay, Diwan S. Rawat. https://doi.org/10.1016/j.ejmech.2020.112276). This recent study and some other studies used and recommended ex vivo assays in order to see the effect of antibiotics on the MRSA bacteria which can be protected inside human cells (Staphylococcus aureus: new evidence for intracellular persistence. Christian Garzoni, William L. Kelley. https://doi.org/10.1016/j.tim.2008.11.005). That is why we employed ex vivo time kill assay in our study.
3. Why did author measured MIC mostly for MRSA clinical isolates and only one MSSA? Ideally, more MSSA isolates should be added. Authors wrote that MRSA is highly infectious, but it is completely wrong. Infectivity of MRSA and MSSA are same. Difference is the presence of mecA on SCCmec to confer beta-lactam resistance, which is not related to activity of MB.
***a. We agreed with the reviewer that the sentence can be misunderstood and modified this sentence by removing these controversial statement (in fact we did not want to make a comparison with MSSA strains regarding infectivity).
***b. In fact, in this study, we aimed to investigate the antimicrobial effects of MB on MRSA isolates. MSSA was used as control and to compare with the resistant MRSA strain in time-kill study. Since MSSA infections can be treated with standard conventional antibiotics (penicillines, cephalosporins, eritromycin , clindamycin cipro, SXT etc… ), the real problem is to treat drug resistant MRSA infections which has poor antibiotic options. Due to delayed proper treatment and less favorable alternative treatment regimens, the mortality rate of MRSA bloodstream infections is double that of equivalent infections attributed to methicillin-susceptible strains and less antibiotic options are available for treatment. (De Kraker M.E.; Davey P.G.; Grundmann H.; BURDEN study group. Mortality and hospital stay associated with resistant Staphylococcus aureus and Escherichia coli bacteremia: estimating the burden of antibiotic resistance in Europe. PLoS. Med. 2011, 8(10), e1001104). So; instead of MSSA treatment, MB can be an alternative for the antibiotics to which MRSA bacteria are resistant.
4. Clinical isolates of MRSA are heterologous population. Some of them must have any drug-efflux pump that is effective also to MB, that may be the reason why MIC values had a wide range. Did author determine any genotype of MRSA? Did author find any correlation with higher or lower MIC to any attributes of isolates (genotype, specimen type, antimicrobial susceptibility, etc.)?
***We agreed with the reviewer and analyzed the isolates in terms of specimen type, antimicrobial susceptibility to different antibiotics. Accordingly, we added new subjects to the discussion. We could not perform genotyping and this point is now added as a limitation of our study in the discussion section of the manuscript.
5. Figure 5, readers cannot see growth of isolates in the left part. Authors should take photo from the side of agar surface, so that readers can see better the growth of bacteria even background is dark.
***These revisions were made in accordance with the reviewer's recommendations. New photos were included in the manuscript.

Reviewer 2 Report
Comments and Suggestions for Authors
Light-activated methylene blue (MB) is frequently used as a photosensitizer in clinical practice. In many other similar studies, methylene blue is considered effective for antimicrobial photodynamic treatment due to its ability to produce reactive oxygen species, which lead to bacterial death. MICs and an in vitro/ex vivo time-killing assay of methylene blue as a single agent were performed on a very small set of clinical MRSA isolates in the study of Gazel et al. The MRSA isolates have very little characteristics. There was no mention of the year of isolation, and there was no mention of susceptibility to various antibiotics. The authors find that the MIC of MB was in the range of 16–64 mg/L. Was there any correlation (synergism, antagonism) with susceptibility to any other antibiotics?
Major comments:
L92. «Twenty-two isolates had an MIC of 16 μg/ml, seven isolates had an MIC value of 16 μg/ml and one isolate’s MIC was 64 μg/ml…» Please correct the sentence.
L98. By using the agar dilution method, what was the clinical isolate's initial minimum inhibitory concentration?
Figures. The plots should be made by another editor. I would recommend tabulating this data and combining, for instance, in vitro versus ex vivo data for the MRSA (1 table) and MSSA (2 table).
The discussion section should be thoroughly shortened; repetitive phrases from the introduction should be avoided. Lines 132–155 could be presented as 2-3 sentences.
The findings are not adequately reflected in the conclusion. MB is generally considered safe for in-vivo use, with doses of 7.5 mg/kg showing no toxic effects in humans. Would this dosage be adequate to treat S. aureus infections from a PK/PD perspective, considering that the author's findings indicated a MIC range of 16–64 mg/L? The authors' statement in L205 regarding the potential use of MB as a monotherapy option for the treatment of staphylococcal infections is called into question by this.
The manuscript should be shortened or even revised into a short communication (if it is suitable for the Journal).
Author Response
Dear Editor and Reviewers (Please see the attachment)
Appended to this letter is my point-by-point response to the comments raised by the reviewers. As you notice, I agreed with most of the comments raised by the reviewers. The revisions were performed in accordance to the recommendations of Reviewer 1 (marked with yellow), Reviewer 2 (marked with green) and Reviewer 3 (marked with turquoise color) in the text. The revisions recommended by multiple reviewers were marked with gray and newly added sections by own decision were marked pink color.
I would like to take this opportunity to express my sincere thanks to the reviewers who identified areas of our manuscript that needed corrections or modification. I would like to thank you for allowing me to modify and submit the manuscript as an original article.
I am looking forward to see my article being published,
Sincerely Yours,
RESPONSES TO REV #2
- Light-activated methylene blue (MB) is frequently used as a photosensitizer in clinical practice. In many other similar studies, methylene blue is considered effective for antimicrobial photodynamic treatment due to its ability to produce reactive oxygen species, which lead to bacterial death. MICs and an in vitro/ex vivo time-killing assay of methylene blue as a single agent were performed on a very small set of clinical MRSA isolates in the study of Gazel et al. The MRSA isolates have very little characteristics. There was no mention of the year of isolation, and there was no mention of susceptibility to various antibiotics. The authors find that the MIC of MB was in the range of 16–64 mg/L. Was there any correlation (synergism, antagonism) with susceptibility to any other antibiotics?
***According to the critics of the reviewer, more MRSA isolates were included in the study to better represent the universe (by approval of the ethical committee and head of dep.). The years of isolation, ages of patients, collected wards, antibiotic susceptibility were included in the manuscript. We could not find any correlation of MIC of MB (synergism, antagonism) with susceptibility to any other antibiotics, however, we employed a new test (antibiotic + MB disc synergy test) in order to analyze the synergy between beta-lactam antibiotics and MB.
EUCAST 2024: Most S. aureus are penicillinase producers and some are methicillin resistant. Either mechanism renders them resistant to benzylpenicillin, phenoxymethylpenicillin, ampicillin, amoxicillin, piperacillin and ticarcillin. Isolates that test susceptible to benzylpenicillin and cefoxitin can be reported susceptible to all penicillins. Isolates that test resistant to benzylpenicillin but susceptible to cefoxitin are susceptible to β-lactam β-lactamase inhibitor combinations, the isoxazolylpenicillins (oxacillin, cloxacillin, dicloxacillin and flucloxacillin) and nafcillin. Isolates that test resistant to cefoxitin are resistant to all penicillins. ("The European Committee on Antimicrobial Susceptibility Testing. Breakpoint tables for interpretation of MICs and zone diameters. Version 14.0, 2024. http://www.eucast.org”)
In the revision stage, a new disc diffusion-MB synergy test was employed to investigate if there is any correlation (synergy or antagonism) with susceptibility to any other antibiotics. In this assay; sub-inhibitory concentrations (sub-MIC) of MB were analyzed since the inhibitory concentrations of MB completely eradicated and inhibited the growing of bacteria. Secondly via this assay; as an advantage; lower and less toxic concentrations of MB were investigated. Here, we tried to understand whether MB (low dose)+penicillin or MB+ cefoxitin combinations could be effective on MRSA isolates or not (just like the effect of ampicillin+sulbactam combination).
- L92. «Twenty-two isolates had an MIC of 16 μg/ml, seven isolates had an MIC value of 16 μg/ml and one isolate’s MIC was 64 μg/ml…» Please correct the sentence. L92.
***This sentence was revised as recommended.
- L98. By using the agar dilution method, what was the clinical isolate's initial minimum inhibitory concentration? L98.
***It was 16 mcg/ml. This datum was added to the manuscript.
- Figures. The plots should be made by another editor. I would recommend tabulating this data and combining, for instance, in vitro versus ex vivo data for the MRSA (1 table) and MSSA (2 table).
***These figures were tabulated as recommended.
- The discussion section should be thoroughly shortened; repetitive phrases from the introduction should be avoided. Lines 132–155 could be presented as 2-3 sentences.
***We agreed with the reviewer and shortened the discussion and introduction section; additionally, repetitive phrases were removed.
- The findings are not adequately reflected in the conclusion. MB is generally considered safe for in-vivo use, with doses of 7.5 mg/kg showing no toxic effects in humans. Would this dosage be adequate to treat S. aureus infections from a PK/PD perspective, considering that the author's findings indicated a MIC range of 16–64 mg/L? The authors' statement in L205 regarding the potential use of MB as a monotherapy option for the treatment of staphylococcal infections is called into question by this.
*** The following contradictory sentence was removed from the text; “As another limitation, doses of ≥7 mg methylene blue/kg may cause methemoglobinemia and methylene blue compounds are safe in lower doses [63]” since doses of 7.5 mg/kg showed no toxic effects in humans as we cited in our manuscript previously.
*** There are a few studies evaluating the plasma/serum levels of MB on healthy volunteers. According to those studies higher concentrations of MB could be obtained in blood when MB is administered orally (Walter-Sack I.; Rengelshausen J.; Oberwittler H.; Burhenne J.; Mueller O.; Meissner P.; Mikus G. High absolute bioavailability of methylene blue given as an aqueous oral formulation. Eur. J. Clin. Pharmacol. 2009, 65(2), 179-89) and (Pharmacokinetics and organ distribution of intravenous and oral methylene blue; Peter et al, DOI: 10.1007/s002280000124). Moreover, animal experiments demonstrated that twentyfold higher concentrations can be obtained in brain after i.v. administration (Peter et al, DOI: 10.1007/s002280000124). In a recent study Di Stefano et al reported that methylene blue AUC0-t was 10.7 ±â€¯6.7 μg/mLxh after 100 mg and 25.2 ±â€¯7.4 μg/mLxh after 200 mg oral administration of methylene blue in tablet form. Only one adverse event was detected as related to methylene blue: a mild increase in alanine aminotransferase. (Di Stefano, https://doi.org/10.1016/j.cct.2018.06.001). In another study, they investigated the safety and bioavailability of methylene blue after single oral doses of 200 and 400 mg in healthy volunteers and reported that MB concentration of 32.94 μg/mL and 38.08 μg/mL were obtained after administration of MB, respectively. Only non-serious adverse events occurred during the study (https://doi.org/10.1016/j.cct.2011.11.006). When these studies are evaluated together we can conclude that MB may have a potential to be used as an antimicrobial drug since 16 and 32 mcg/ml MICs were detected in our study. Furthermore, we observed a significant synergy between cefoxitin and methylene blue at a concentration of 1-2 mcg/ml in 20% of the strains (susceptibility development). So, we evaluated that MB can also be used at lower/sub-inhibitory doses as a combination compound for sensitization of beta lactam antibiotics.
MB can also be used in the topical form. There are studies reporting the effects of MB at a concentration of 500 mcg/ml for treatment of oral candidiasis (Treatment of oral candidiasis with methylene blue-mediated photodynamic therapy in an immunodeficient murine model- DOI: 10.1067/moe.2002.120051). Furthermore, MB collutory preparations at a concentration of 10000 mcg/ml had been licensed in Turkey for treatment of oral candidiasis (https://www.tabilac.com/en/category/drugs/buco-bleu). So, higher concentrations of MB can be achieved in the targeted superficial tissues and these collutory, gel or cream forms of MB can be used to treat mucosal and skin infections of MRSA without any toxic effect. They can be absorbed percutaneously in sufficient amounts to cause antimicrobial effect on skin and mucosa but insufficient to cause systemic effects (Liposomal methylene blue hydrogel for selective photodynamic therapy of acne vulgaris. Journal of drugs in dermatology : JDD, 8(11), 983–990) and (Methylene blue mediated antimicrobial photodynamic therapy in clinical human studies: The state of the art. Rebeca Boltes Cecatto et al; https://doi.org/10.1016/j.pdpdt.2020.101828). Additionally, mupirocin antibiotic is topically used for nasal decolonization of S. aureus and MB can be an alternative to this drug since higher concentrations of MB can be applied on the surface of nasal mucosa. As another claim, MB could be useful for treatment of severe neurological infections such as MRSA meningitis since much higher concentrations of MB can be achieved in the central nervous system than in blood (Peter et al, DOI: 10.1007/s002280000124)
Statements and speculations discussing the potential use of MB as a monotherapy option for the treatment of staphylococcal infections were added to the discussion, accordingly. Also we discussed the synergistic effect since we added a new antibiotic + sub-MIC MB synergy assay to the study.
- The manuscript should be shortened or even revised into a short communication (if it is suitable for the Journal).
We shortened the article; especially the repetitive phrases were removed. A new antibiotic-MB synergy assay analyzing the antimicrobial effects of MB in combination with beta-lactam antibiotics (at sub-inhibitory doses of MB compound). In order to be used in our yearly academicals performance score, we kindly ask you if it is possible to evaluate as an original research article.

Reviewer 3 Report
Comments and Suggestions for Authors
1. The author's mentioned "ex vivo" experiments involve co-culturing isolates with a cell line. The reason for using an osteosarcoma cell line is unclear. Moreover, this process merely performs a time-kill assay in the background of cell culture medium, which does not simulate conditions within a living organism.
2. In the ex-vivo experiment, at a concentration of 2 mcg/ml, bacteria briefly regrow before ceasing growth again. Would the author discuss this phenomenon?
3. Based on the experimental results, whether the required concentration of MB for effective antibacterial action is safe remains uncertain. While the author touches upon safety considerations in the discussion, no definitive conclusions or speculations are provided.
4. Apart from the MIC, the experiment was conducted solely on a single bacterial strain. Why weren't experiments conducted on bacterial strains with different MICs?
5. The author can utilize molecular typing to understand the background of bacterial strains, which would be helpful in assessing the response of different strains to the drug.
6. The presentation style of Figures 1-4 may result in certain data being obscured. It is suggested to change it to a 2D format.
7. The scale of this study is too limited. If it is to be published, it is recommended to consider presenting it as a "Communication."
Author Response
Dear Editor and Reviewers (Also see attachment)
Appended to this letter is my point-by-point response to the comments raised by the reviewers. As you notice, I agreed with most of the comments raised by the reviewers. The revisions were performed in accordance to the recommendations of Reviewer 1 (marked with yellow), Reviewer 2 (marked with green) and Reviewer 3 (marked with turquoise color) in the text. The revisions recommended by multiple reviewers were marked with gray and newly added sections by own decision were marked pink color.
I would like to take this opportunity to express my sincere thanks to the reviewers who identified areas of our manuscript that needed corrections or modification. I would like to thank you for allowing me to modify and submit the manuscript as an original article.
I am looking forward to see my article being published,
Sincerely Yours,
RESPONSES TO REV #3
- The author's mentioned "ex vivo" experiments involve co-culturing isolates with a cell line. The reason for using an osteosarcoma cell line is unclear. Moreover, this process merely performs a time-kill assay in the background of cell culture medium, which does not simulate conditions within a living organism.
***a. U2OS is a osteosarcoma cell line with epithelial morphology that was derived from a moderately differentiated sarcoma of the tibia of a osteosarcoma patient (1,2). It contains wild type p53+/+ and pRb+/+ tumor suppressor genes and gives similarities like normal cells. Moreover as a cancer cell line it has immunoresistance and ability to survive in disfavorable conditions like starved condition or infections. It is widely used for its efficiency of nucleic acids (DNA or RNA) in researches of transfection (3) and viral infection (4), as well as bacterial infections (5). In addition S. aureus which is recently considered considered to be an extracellular pathogen can be internalized in epithelial cells (6) which has similarities with osteosarcoma cells.
Some references:
1) https://www.atcc.org/products/htb-96#:~:text=U%2D2%20OS%20is%20a,%2C%20White%2C%20female%20osteosarcoma%20patient.
2) https://www.sciencedirect.com/science/article/abs/pii/B9780123748959000281?via%3Dihub
3) https://doi.org/10.1016/j.prp.2023.154948
4) https://pubmed.ncbi.nlm.nih.gov/23659165/
5) https://www.nature.com/articles/s41467-020-18857-z
6) https://www.ncbi.nlm.nih.gov/pmc/articles/PMC107895/
***b. Recently, it has been revealed that S. aureus, which was considered to be an extracellular pathogen, can invade and survive inside mammalian cells. These infected cells may provide a reservoir and protective niche for S. aureus, and thus, the bacterial cells may be relatively protected from conventional antibiotics. (Synthesis of novel monocarbonyl curcuminoids, evaluation of their efficacy against MRSA, including ex vivo infection model and their mechanistic studies. Gagandeep, Prince Kumar, Shamseer Kulangara Kandi, Kasturi Mukhopadhyay, Diwan S. Rawat. https://doi.org/10.1016/j.ejmech.2020.112276). Recent studies used and recommended to use ex vivo human cell lines in order to see the effect of antibiotics on the MRSA bacteria which can be protected inside human cells (Staphylococcus aureus: new evidence for intracellular persistence. Christian Garzoni, William L. Kelley. https://doi.org/10.1016/j.tim.2008.11.005, https://doi.org/10.1016/j.ejmech.2020.112276). That is why we employed ex vivo time kill assay in order to see the intracellular killing effect of the MB our study.
***c. Ex-vivo tests can partially mimicry in vivo tissue environment and they are superior to the in vitro tests since the additional effects of the human cells on bacteria can be observed in combination with antimicrobial drug. It has been suggested that ex vivo models may be more clinically relevant, since in vitro studies do not take into account host proteins that can neutralize antiseptic/antimicrobial activity (Lepelletier D, Maillard JY, Pozzetto B, Simon A. Povidone Iodine: Properties, Mechanisms of Action, and Role in Infection Control and Staphylococcus aureus Decolonization. Antimicrob Agents Chemother. 2020 Aug 20;64(9):e00682-20. doi: 10.1128/AAC.00682-20) and (Ex vivo porcine vaginal mucosal model of infection for determining effectiveness and toxicity of antiseptics. J Appl Microbiol 115:679–688. doi: 10.1111/jam.12277).
- In the ex-vivo experiment, at a concentration of 2 mcg/ml, bacteria briefly regrow before ceasing growth again. Would the author discuss this phenomenon?
***In low concentrations sometimes we can see this phenomenon. It could be due to the unstable effect of the antimicrobial at very low concentrations. Some examples to this phenomenon: https://www.creative-biolabs.com/drug-discovery/therapeutics/time-kill-analysis-for-fungi.htm
Kwon et al. AMB Expr (2019) 9:122 https://doi.org/10.1186/s13568-019-0843-0 (Figure 3c in the link)
- Based on the experimental results, whether the required concentration of MB for effective antibacterial action is safe remains uncertain. While the author touches upon safety considerations in the discussion, no definitive conclusions or speculations are provided.
*** There are a few studies evaluating the plasma/serum levels of MB on healthy volunteers. According to those studies higher concentrations of MB could be obtained in blood when MB was administered orally (Walter-Sack I.; Rengelshausen J.; Oberwittler H.; Burhenne J.; Mueller O.; Meissner P.; Mikus G. High absolute bioavailability of methylene blue given as an aqueous oral formulation. Eur. J. Clin. Pharmacol. 2009, 65(2), 179-89) and (Pharmacokinetics and organ distribution of intravenous and oral methylene blue; Peter et al, DOI: 10.1007/s002280000124). Moreover, animal experiments demonstrated that tenfold higher concentrations can be obtained in brain after i.v. administration (Peter et al, DOI: 10.1007/s002280000124). In a recent study Di Stefano et al reported that methylene blue AUC0-t was 10.7 ±â€¯6.7 μg/mLxh after 100 mg and 25.2 ±â€¯7.4 μg/mLxh after 200 mg oral administration of methylene blue in tablet form. Only one adverse event was detected as related to methylene blue: a mild increase in alanine aminotransferase. (Di Stefano, https://doi.org/10.1016/j.cct.2018.06.001). In another study, they investigated the safety and bioavailability of methylene blue after single oral doses of 200 and 400 mg in healthy volunteers and reported that MB concentration of 32.94 μg/mL and 38.08 μg/mL were obtained after administration of MB, respectively. Only non-serious adverse events occurred during the study (https://doi.org/10.1016/j.cct.2011.11.006). When these studies are evaluated together we can conclude that MB may have a potential to be used as an antimicrobial drug since 16 and 32 mcg/ml MICs were detected in our study. Furthermore, we performed a new synergy test and we observed a significant synergy between cefoxitin and methylene blue at a concentration of 2 mcg/ml in 20% of the strains (susceptibility development) while weaker synergy was observed for all other isolates. So, we evaluated that MB can also be used at lower/sub-inhibitory doses as a combination compound for sensitization of bacteria to beta lactam antibiotics since isolates that test resistant to benzylpenicillin but susceptible to cefoxitin are also susceptible to β-lactam β-lactamase inhibitor combinations, the isoxazolylpenicillins and nafcillin (EUCAST 2024).
MB can also be used in the topical form. There are studies reporting the effects of MB at a concentration of 500 mcg/ml for treatment of oral candidiasis (Treatment of oral candidiasis with methylene blue-mediated photodynamic therapy in an immunodeficient murine model- DOI: 10.1067/moe.2002.120051). Furthermore, MB collutory preparations at a concentration of 10000 mcg/ml had been licensed in Turkey for treatment of oral candidiasis (https://www.tabilac.com/en/category/drugs/buco-bleu). So, higher concentrations of MB can be achieved in the targeted superficial tissues and these collutory, gel or cream forms of MB can be used to treat mucosal and skin infections of MRSA without any toxic effect. They can be absorbed percutaneously in sufficient amounts to cause antimicrobial effect on skin and mucosa but insufficient to cause systemic effects (Liposomal methylene blue hydrogel for selective photodynamic therapy of acne vulgaris. Journal of drugs in dermatology : JDD, 8(11), 983–990) and (Methylene blue mediated antimicrobial photodynamic therapy in clinical human studies: The state of the art. Rebeca Boltes Cecatto et al; https://doi.org/10.1016/j.pdpdt.2020.101828).
Additionally, mupirocin antibiotic is topically used for nasal decolonization of S. aureus and MB can be an alternative to this drug since higher concentrations of MB can be applied on the surface of nasal mucosa. As another claim, MB could be useful for treatment of severe neurological infections such as MRSA meningitis since much higher concentrations of MB can be achieved in the central nervous system than in blood (Peter et al, DOI: 10.1007/s002280000124). New statements and speculations were added to the discussion, accordingly.
- Apart from the MIC, the experiment was conducted solely on a single bacterial strain. Why weren't experiments conducted on bacterial strains with different MICs?
***Since, we have to use dozens of agar plates for only one strain in the time kill assay, we limited the test with a standard bacterial strain and a clinical isolate. However, we added a new antibiotic-MB synergy assay to analyze the antimicrobial effects of MB in combination with beta-lactam antibiotics (at sub-inhibitory doses of MB compound) on 10 MRSA isolates from different wards and with different antimicrobial resistance profiles.
- The author can utilize molecular typing to understand the background of bacterial strains, which would be helpful in assessing the response of different strains to the drug
***We agreed with the reviewer and analyzed the isolates in terms of specimen type, antimicrobial susceptibility to different antibiotics. However, we could not perform genotyping and this point is added as a limitation of our study in the discussion section of the manuscript. We planned to perform genotyping on isolates as a separate project.
- The presentation style of Figures 1-4 may result in certain data being obscured. It is suggested to change it to a 2D format.
***According to the suggestions of both 2 reviewers, the figures were converted to table format.
- The scale of this study is too limited. If it is to be published, it is recommended to consider presenting it as a "Communication."
***We increased the scale of the adding 74 new isolates and performing a new antibiotic-MB synergy assay to analyze the antimicrobial effects of MB in combination with beta-lactam antibiotics (at sub-inhibitory doses of MB compound). In order to be used in our yearly academicals performance score (where communications are excluded), we kindly ask you if it is possible to evaluate as an original research article.

Round 2
Reviewer 1 Report
Comments and Suggestions for Authors
Now the revised version is well written and understandable.
Author Response
Thank you very much for accepting my article.
Reviewer 2 Report
Comments and Suggestions for Authors
The manuscript was corrected; the results and discussion were revised. The manuscript has been improved with additional results.
Author Response
Thank you very much for accepting my article
Reviewer 3 Report
Comments and Suggestions for Authors
The presentation style of Figures 1-4 may result in certain data being obscured. It is suggested to change it to a 2D format.
Author Response
Dear Editor
Appended to this letter is my point-by-point response to the questions raised by the academic editor and reviewer. As you notice, I agreed with most of the comments raised by the editor and reviewer. All the revisions were marked with yellow in the text. I would like to take this opportunity to express my sincere thanks to the reviewers who identified areas of our manuscript that needed corrections or modification. I would like to thank you for allowing me to modify and submit the manuscript as an original article.
I am looking forward to see my article being published,
Sincerely Yours,
Response to the Reviewer #3
- The presentation style of Figures 1-4 may result in certain data being obscured. It is suggested to change it to a 2D format.
* These graphs were changed to 2D format as suggested by the reviewer
